# Microphysics parameterization sensitivity of the WRF Model version 3.1.7 to extreme precipitation: evaluation of the 1997 New Year's flood of California

Elcin Tan

Department of Meteorological Engineering, Istanbul Technical University, Istanbul, 34469, Turkey *Correspondence to*: Elcin Tan (elcin.tan@itu.edu.tr)

**Abstract.** Providing high accuracy in quantitative extreme precipitation forecasting (QEPF) is still a challenge. California is vulnerable to extreme precipitation, which occurs due to atmospheric rivers and might be more intense with climate change. Accordingly, this study is an attempt to evaluate the

- extreme precipitation forecasting performance of a QPF model, the Weather Research and Forecast Model, version 3.1.7, for the extreme precipitation event that caused the 1997 New Year's flood in California. Sensitivities of 19 microphysics schemes are tested by utilizing 18 various Goodness of Fit (GoF) tests for hourly and point-wise comparisons between 3-km horizontal domain resolution simulations of the WRF Model and observations. The results indicate that the coefficient of persistence
- (cp) is the first metric that needs to be evaluated because it determines whether simulation versus observation values are reasonable. Comparisons of 3 out of 8 stations in the American River Watershed passed this test. The results also show that Normalized Root Mean Square Errors (NRMSE) and Percent Bias (PBIAS) metrics are more representative than others due to their ability to discriminate model performances. Further, microphysics (MP) schemes are also significantly sensitive to location.
- Although 3 of the stations that passed the cp test are quite near to each other spatially, different MP schemes become prominent for different observation locations. For instance, for the ALP station, MP3, MP8, MP17, and MP28 indicate better performances, whereas the errors of MP3, MP8, MP9, and MP17 are less than other MPs for the BTA station. However, MP11 has the only reasonable results, according to cp values for the CAP station. The MPs are also evaluated for 72-hr and basin-averaged
- precipitation estimations of the WRF Model by means of true percent relative errors. The results show that the accuracy of the WRF Model is much higher for the 72-hr total basin-averaged evaluations than

for the hourly and point-wise comparisons. Thus, the Thompson Scheme (MP8) indicates more trustworthy results than others, with a 3.1% true percent relative error. Although WRF simulations overestimate the 72-hr basin-averaged precipitation for most of the MP schemes, this may not be pronounced for moderate, heavy, and extreme precipitation when hourly and point-wise evaluations are

5 performed but is valid for light precipitation.

## **1** Introduction

The IPCC noted that the magnitude, frequency, and duration of extreme events may increase as a result of climate change (2012). This impact could be significant for regions such as California where drought intensities also tend to increase and excessive amounts of precipitation should therefore be saved for

- future periods of drought. When drought causes these regions to take one step back, extreme precipitation events can take them two steps forward. Hence, smart and new water management strategies should be developed to advance climate change adaptation. Some of the solutions might be related to improving the time and space problem of quantitative extreme precipitation forecasting (QEPF). The Weather Research and Forecasting Model (WRF) (Skamarock et al., 2008) has been
- widely used to increase resilience to extreme precipitation events in California (Jankov et al., 2007; Jankov et al., 2009; Tan, 2010; Eiserloh and Chiao, 2014), which are closely correlated with atmospheric river events (Tan, 2010). Atmospheric rivers are also expected to increase in terms of both frequency and intensity with climate change (Dettinger, 2011). Generally, for precipitation simulations, a combination of the microphysical, cumulus parameterization and the planetary boundary layer
- parameterization schemes is mainly evaluated using the WRF Model, although all of the parameterization schemes and processes have effects on precipitation occurrence. Moreover, microphysical schemes play a significant role if the horizontal resolution of the interested domain is finer than 9 km. Recent studies have been conducted for the evaluation of the microphysics schemes of the WRF Model with respect to winter precipitation in California, with a focus on atmospheric river
- phenomena (Jankov et al., 2007; Jankov et al., 2009; Tan, 2010; Jankov et al., 2011; Han et al., 2013), which is the reason for the extreme precipitation that occurs along the West coast of the US (Zhu and Newell, 1998; Ralph et al., 2004; Tan, 2010; Houze, 2012). Jankov et al. (2007) evaluated the Lin,

Ferrier, WSM6, and Thompson schemes of version 2 of the WRF-ARW for the February 27, 2006, atmospheric river event by using a 3-km horizontal grid resolution. Jankov et al. (2009) also investigated the Lin, WRF Single-Moment 6-Class (WSM6), Thompson, and Schultz microphysics schemes for a 3-km horizontal resolution by simulating December 30–31, 2005, January 1, 2006,

- February 1, 2006, February 27, 2006, and March 5, 2006 events. Jankov et al. (2011) included the double-moment Morrison microphysics scheme in the four MP physics they used in their previous research (Jankov et al., 2009) and changed the finest horizontal grid resolution to 4 km to evaluate only the December 30–31, 2005 event, using version 3.0 of the WRF Model. Han et al. (2013) evaluated the Goddard scheme, WRF single-moment 6-class scheme, Thompson scheme and Morrison double-
- moment scheme of version 3.1 of the WRF Model by simulating the December 30–31, 2005 atmospheric river (AR) event that occurred in northern California and Nevada, with a 1.3-km horizontal resolution. Tan (2010) calibrated the WRF Model version 3.1.1 by using the Kessler, Lin, WSM 3-class, WSM 5-class, Ferrier, WSM 6-class, Goddard GCE, Thompson, Morrison, WDM 5-class, WDM 6-class, and Old Thompson schemes for the 1997 New Year's event at the American River Watershed,
- with a 3-km horizontal resolution and 72-hr basin-averaged consideration for the purpose of probable maximum precipitation (PMP) estimation. She also validated the WRF Model using the Thompson microphysics scheme for 42 historical atmospheric river events that occurred between 1951 and 2002. Hereupon, the main goal of this study is to evaluate hourly quantitative extreme precipitation forecasting (QEPF) performance of the WRF Model for the most extreme atmospheric river event in the
- history of California. The 1997 New Year's flood, which occurred between December 26, 1996, and January 3, 1997, has been ranked as the fifth of California's top 15 weather events in the 1900s (URL-1) and is still the major impactful atmospheric river event of the state. Thus, the 1997 New Year's atmospheric river event is simulated for the evaluation of version 3.7.1 of the WRF Model. In this way, new MP schemes that were not evaluated by Tan (2010) are also included in this study. As a result,
- hourly QPF of the total 19 bulk parameterization schemes is evaluated by using 18 different Goodness of Fit (GoF) tests at a 3-km horizontal grid resolution by comparing hourly ground-based rain gauge observations. Additionally, 72-hr basin-averaged values are also discussed for comparison with the previous studies.

This study is organized as follows: Section 2 presents data used for the simulations and comparisons. The model design is detailed in Section 3, including MP schemes and evaluation methods. Section 4 discusses results in terms of both in-situ observations and 72-hr basin-averaged values. Finally, the study is concluded in Section 5.

5

## 2 Data

# 2.1 Reanalysis Data Set

WRF simulations are initialized by using a 6-hourly NCEP/NCAR Global Reanalysis Model (Kalnay et al., 1996). The resolution of this model is T62 (209 km) with 28 vertical sigma levels. The dataset
(2.5°x2.5°) is continuously available online, starting from 1948, at the Research Data Archive (RDA) of the University Corporation for Atmospheric Research (UCAR) (NCEP et al., 1994).

#### 2.2 Observational Data

Hourly rain-gauge data are obtained from the California Data Exchange Center (CDEC) (URL-2) for the American River Watershed where Pacific Standard Time (PST) is utilized as the local time. Because

- 15 the WRF Model uses Coordinated Universal Time (UTC), all the comparisons are scaled accordingly. That is, the WRF Model comparisons are presented in this study based on UTC and 72-hr basinaveraged comparisons are kept with local time comparisons in order to be consistent with the literature. Hourly rainfall rates of the WRF model are evaluated by considering 8 rain gauge stations: Alpha (ALP- 2316 m), Beta (BTA- 2316 m), Owens Camp (OWC- 1371 m), Caples Lake (CAP- 2438 m),
- Hell Hole (HLH- 1396 m), Pilot Hill (PIH- 366 m), Lincoln (LCN- 61 m), and Hurley (HUR- 11 m) (Figure 1). The stations' IDs and their elevations are given in the parentheses.
  Precipitation rates are classified as follows: Light: Trace-2.5 mm/hr, Moderate: 2.5mm/hr-7.5mm/hr, Heavy: 7.5mm/hr-10mm/hr, and Extreme: >10mm/hr.

The hourly time series of each station shows that the ALP and BTA stations observed extreme 25 precipitation during the New Year's flood of 1997 (Figure 2).

5

Figure 1: The map of the American River Watershed and the precipitation stations