# Peer review of "Microphysics parameterization sensitivity of the WRF Model version 3.1.7 to extreme precipitation: evaluation of the 1997 New Year's flood of California"

_Geoscientific Model Development, 2016_

## Referee Comment (RC1) · Anonymous Referee #1 · 29 Jul 2016

The manuscript presents an evaluation study of WRF simulated precipitation using some surface observations. The five-day WRF simulations using 19 different cloud microphysical schemes are compared with surface precipitation measurements over 8 stations in California. As a pure model comparison study which is only based on a five-day precipitation event, I feel the spatial and temporal scales are too small in the experiment design to judge different cloud microphysical schemes, and the analysis in this study is inadequate to advance our understanding in model's physics. Specifically, the manuscript lacks a conclusive assessment of the different MPs, as they behave quite diverse at different sites. Moreover, the simple model-observation comparisons

without any process-level diagnostics in this study fail to shed any light on the essential differences between MPs. Therefore, I cannot recommend publication of this manuscript in GMD.

———————————————————

---

## Referee Comment (RC2) · Anonymous Referee #2 · 8 Aug 2016

Comments to the Author
Review of 'Microphysics parameterization sensitivity of the WRF Model version 3.1.7 to extreme precipitation: evaluation of the 1997 New Year's flood of California' by Elein Tan

This study tried to evaluate the WRF model performance of microphysical schemes on predicting an extreme precipitation event by using 18 Goodness of Fit tests for hourly and point-wise and 72-hr basin-averaged comparison between observations and model outputs. While the attempt to evaluate 19 available bulk microphysical schemes of the WRF model of the current version is of merit, there are two major deficiencies in this paper. Firstly, this study used only 3 stations of rain gauge data for the GoF tests, which can hardly lead to any statistically meaningful result. Secondly, the method used in this study is not suitable for evaluating the performance of microphysical schemes because all the metrics in the GoF tests are the final scores about the performance, which cannot show us what is wrong with the schemes on a process level and thus failed to tell us how to improve the schemes, which is an important aspect for an evaluation paper. Therefore, this reviewer has to suggest the editor to reject the publication of the manuscript in GMD.

---

## Referee Comment (RC3) · Anonymous Referee #3 · 22 Aug 2016

This paper has potential to be a very useful contribution to the use of high-resolution NWP models, in particular for the forecasting of extreme precipitation events. It is well-written and clearly presented and oviously represents a great deal of work.

However, to be publishable I think the paper requires some further work. The study shows which microphysics (MP) schemes perform the best for this particular event. A key point (and one which would make this paper original, and significant) would be to find the common factors among the "successful" MP schemes, in other words, Why do these particular schemes perform the best? This "Why" is not addressed in

the paper. Without this cause-and-effect relationship being established, it is difficult to know whether the current findings are applicable to other extreme events. Thus, I recommend a major revision.

Minor comments. At the beginining of the manuscript, the model simulations are described as having 28 levels. Later in the manuscript this appears to be 41 levels. Which is true? In either case, is this number of vertical levels sufficient to model events of this type? I think a few sentences here would be useful.

P22, L25ff. "Model uncertainties..." I think the author is rather opening a "can of worms" here. It is not evident whether differences in model results associated with differing MP schemes are comparable with, or greater than, the differences in, say, horizontal resolution. This would represent a significant undertaking, however, with many degrees of freedom. I wonder if using WRF in some idealised framework may be one way to deal with this.

---

## Author Comment (AC1) · 3 Oct 2016

I would like to thank Anonymous Referee #1 for their time to evaluate my manuscript. Here is my replies, following Referee #1's concerns in bold-italics.

***"I feel the spatial and temporal scales are too small in the experiment design to judge different cloud microphysical schemes".***

As I discussed in the summary and conclusion part of this manuscript, precipitation results can vary depending on the domain selection and it needs to be discussed in details to quantify the uncertainty added to the solutions by domain choice. Although domain selection is not a focus on this study, I understand your concern about experiment design. Therefore, I have run the model with a new and bigger domain, as you suggested, to prove that the final results does not change much and the current choice of nested domains are not too small. Although it is suggested that the domain grid numbers should not be less than 50x50 for each nested domains, some people even think this should be at least 100x100, according to my experience this choice really depends on the situation. Moreover, I had tried several different domain choices, not shown in this paper, and this domain was one of the optimum ones.

The new domain configuration (Big domain, hereafter) has been presented in Figure S.1. Its grid points are 121x107, 115x109, 112x106 for domains d01, d02, and d03, respectively. Outer most domain is d01 (black borders), the middle one is d02 (white borders), and the innermost domain is d03 (red borders). The runs took approximately 5 times longer than the runs of the domain presented in the manuscript (Small domain, hereafter).

[Figure]

Figure S1. New domain setup (Big Domain)

The results of only MP1, MP2, MP3, MP4, MP5, MP6, MP7, MP8, and MP28 are given in the Table S.1. due to lack of time and they indicate that having bigger domain setup did not solve the problem that Cp values are still negative for CAP, HLH, HUR, LCN, OWC, and PIH stations. On the other hand, negative Cp values of MP1, MP2, MP4, MP6 for BTA stations are improved to positive Cp values in use of the big domain, as we might expect. PBIAS values of the big domain proves that bigger domain is not necessarily better than the smaller domain. As it is seen from the comparisons of ALP and BTA stations with 9 MP options of the WRF Model, PBIAS values are having more bias when the domain size increases, except MP28.

**Table S.1 Comparison of small and big domain with respect to 8 observation stations**

**Cp Values**

| Domain | ALP Small | ALP Big | BTA Small | BTA Big | CAP Small | CAP Big | HLH Small | HLH Big | HUR Small | HUR Big | LCN Small | LCN Big | OWC Small | OWC Big | PIH Small | PIH Big |
|---|---|---|---|---|---|---|---|---|---|---|---|---|---|---|---|---|
| MP1 | 0.12 | 0.26 | -0.19 | 0.19 | -3.32 | -4.15 | -4.14 | -3.97 | -6.05 | -0.66 | -2369.21 | -2934.69 | -2.19 | -0.81 | -4.78 | -2.68 |
| MP2 | 0.05 | 0.37 | -0.13 | 0.3 | -2.66 | -2.49 | -5.48 | -5.21 | -5.82 | -3.81 | -2530.11 | -3304.35 | -2.28 | -0.83 | -4.89 | -2.81 |
| MP3 | 0.24 | 0.32 | 0.09 | 0.21 | -1.35 | -2.83 | -2.15 | -1.91 | -3.28 | -2.34 | -1490.32 | -1802.32 | -3.84 | -1.22 | -3.8 | -4.84 |
| MP4 | 0.04 | 0.22 | -0.15 | 0.16 | -1.51 | -2.02 | -2.05 | -2 | -4.72 | -4.16 | -1137.02 | -1577.65 | -6.62 | -2.54 | -5.57 | -5.43 |
| MP5 | 0.23 | 0.33 | 0.04 | 0.28 | -2.24 | -3.47 | -2.15 | -2.3 | -4.31 | -2.81 | -1078.18 | -1080.55 | -2.47 | -0.71 | -3.96 | -4.02 |
| MP6 | 0.07 | 0.27 | -0.11 | 0.16 | -2.5 | -3.24 | -3.06 | -3.01 | -3.68 | -3.4 | -1666.06 | -2146.91 | -3.34 | -1.82 | -4.06 | -3.18 |
| MP7 | 0.08 | 0.37 | 0 | 0.37 | -0.64 | -0.95 | -2.03 | -1.18 | -3.51 | -3.23 | -917.13 | -1117.38 | -3.09 | -1.07 | -3.33 | -3.32 |
| MP8 | 0.28 | 0.42 | 0.13 | 0.39 | -0.88 | -1.94 | -2.03 | -1.22 | -3.18 | -2.06 | -1061.81 | -1504.13 | -2.82 | -1.07 | -2.98 | -3.33 |
| MP9 | 0.26 | 0.05 | 0.34 | 0.05 | -1.53 | -2.05 | -2.78 | -2.7 | -1.49 | -1.94 | -1794.71 | -1821.73 | -1.56 | -2.34 | -3.22 | -4.61 |
| MP28 | 0.28 | 0.29 | 0.3 | 0.15 | -1.17 | -1.81 | -1.84 | -2.47 | -1.67 | -1.28 | -1822.81 | -1710.74 | -1.36 | -2.42 | -2.97 | -5.06 |

**PBIAS**

| Domain | ALP Small | ALP Big | BTA Small | BTA Big | CAP Small | CAP Big | HLH Small | HLH Big | HUR Small | HUR Big | LCN Small | LCN Big | OWC Small | OWC Big | PIH Small | PIH Big |
|---|---|---|---|---|---|---|---|---|---|---|---|---|---|---|---|---|
| MP1 | 28.7 | 68.3 | 20.9 | 52.2 | 130.4 | 174.2 | 91 | 131.4 | 74.9 | 20.8 | 12016.1 | 14213.5 | 20.8 | 32.6 | 129.1 | 140.9 |
| MP2 | 45.8 | 67.1 | 35 | 48.4 | 115 | 130.6 | 136.7 | 161.6 | 121.3 | 136.5 | 13139.5 | 15731.1 | 33.9 | 34.1 | 142 | 151.9 |
| MP3 | 4.6 | 54.8 | -1.3 | 39.4 | 102.2 | 158.7 | 78.5 | 93.2 | 96.5 | 97.6 | 10493.1 | 11991 | 40 | 25.9 | 129.1 | 185.8 |
| MP4 | 39.9 | 59.0 | 27.6 | 41.3 | 104.3 | 140.1 | 76.9 | 97.3 | 123.7 | 165.7 | 8905.1 | 10700.9 | 81.8 | 67.7 | 182.3 | 220.3 |
| MP5 | -22.5 | 22.0 | -19.9 | 24.5 | 136.6 | 186.3 | 78.9 | 90.7 | 99.3 | 97.8 | 8324 | 8974.2 | 16.3 | -4.6 | 138.2 | 190.3 |
| MP6 | 35.5 | 57.2 | 25 | 40.7 | 129 | 158.7 | 98.6 | 115.5 | 101.1 | 148.1 | 10525.1 | 12708.1 | 50.9 | 49.8 | 148.3 | 168 |
| MP7 | 51.3 | 43.9 | 38.8 | 30.2 | 69.9 | 110.6 | 58.7 | 73 | 96.9 | 132.4 | 7864.8 | 8863.5 | 58.7 | 42.4 | 156.5 | 174.5 |
| MP8 | 2.9 | 37.6 | -4.7 | 21.2 | 71.7 | 130.8 | 57.4 | 82.1 | 83.2 | 103.2 | 8016.9 | 10600.4 | 30.5 | 38.5 | 116.8 | 170.7 |
| MP9 | 63.2 | 75.1 | 48.3 | 58.4 | 135.4 | 130.4 | 129 | 112.7 | 64.5 | 42 | 11870.1 | 10621.3 | 65.6 | 55.4 | 192.9 | 210.8 |
| MP28 | 36.1 | 28.3 | 22.2 | 11.5 | 106.3 | 114.7 | 99 | 102.8 | 81.2 | 31.6 | 12189.8 | 11091.1 | 52.8 | 40.5 | 179.9 | 204.8 |

**NRMSE**

| Domain | ALP Small | ALP Big | BTA Small | BTA Big | CAP Small | CAP Big | HLH Small | HLH Big | HUR Small | HUR Big | LCN Small | LCN Big | OWC Small | OWC Big | PIH Small | PIH Big |
|---|---|---|---|---|---|---|---|---|---|---|---|---|---|---|---|---|
| MP1 | 117.1 | 107.5 | 131.3 | 108.2 | 218 | 238 | 165.5 | 162.7 | 235 | 114.2 | 6975 | 7762.6 | 126.3 | 95.1 | 217 | 173.1 |
| MP2 | 122.2 | 99.8 | 128.8 | 102.1 | 200.7 | 195.9 | 186.2 | 182.4 | 231.3 | 194.1 | 7211.6 | 8239.4 | 128.1 | 96 | 219.4 | 176.4 |
| MP3 | 109.1 | 102.9 | 114.6 | 107.6 | 160.8 | 205.3 | 129.7 | 125 | 183.2 | 161.7 | 5535.9 | 6085.2 | 155.5 | 105.5 | 198 | 218.8 |
| MP4 | 122.4 | 110.4 | 128.6 | 110 | 166 | 182.2 | 127.4 | 126.6 | 211.7 | 201.2 | 4834.8 | 5694.1 | 195.1 | 133 | 231.9 | 229.7 |
| MP5 | 109.5 | 102.5 | 118 | 102.5 | 188.9 | 221.9 | 129.7 | 132.7 | 204.1 | 172.8 | 4707 | 4711.8 | 131.6 | 92.5 | 201.2 | 202.6 |
| MP6 | 120.4 | 106.3 | 126.5 | 110.2 | 196.3 | 216 | 147.3 | 146.3 | 191.6 | 185.7 | 5852.7 | 6642.3 | 147.3 | 118.8 | 203.4 | 184.9 |
| MP7 | 119.8 | 98.8 | 120.3 | 95.6 | 134.1 | 146.5 | 127 | 107.8 | 187.9 | 182 | 4341.6 | 4791.9 | 142.9 | 101.8 | 187.9 | 187.9 |
| MP8 | 105.7 | 95.2 | 112.1 | 94 | 120.3 | 179.8 | 126.9 | 108.6 | 181 | 154.9 | 4670.7 | 5558.3 | 138.2 | 101.7 | 180.2 | 187.8 |
| MP9 | 107.6 | 121.7 | 97.2 | 116.9 | 166.7 | 183.1 | 141.9 | 140.4 | 139.7 | 151.8 | 6071.2 | 6116.7 | 113.2 | 129.2 | 185.4 | 213.9 |
| MP28 | 105.6 | 104.9 | 100.8 | 111 | 154.4 | 175.6 | 123 | 136 | 144.5 | 133.8 | 6119.6 | 5928.5 | 108.6 | 130.8 | 180.3 | 222.5 |

For ALP and BTA comparisons, NRSMEs are reduced in big domain, except MP28, again. When we compare increase ratio in PBIAS and decrease ratio in NRSME, PBIAS increases about 330%, whereas NRSME reduces about 11% with bigger domain. Therefore, I prefer to use smaller domain with much less PBIAS.

Related with your concern about the temporal scale, I do not think that the temporal scale is too small. If we consider that the performance of weather prediction models is getting worse after 5 days and they cannot be reliable after 7 days, 5-day simulations would be enough for time scales. If you mean that the starting day of the simulation is too early, you would be right if I used operational initial and boundary conditions. Since I use re-analysis data and this is some kind of hindcast analysis, I do not think that the results would change much. As I also discussed in the conclusion section, initial time setup of the event of interest might be an significant source of error for operational purposes, which is also out of scope for this manuscript.

Table 2. 72-hr Total Basin Averaged Precipitation Comparisons for Small and Big Domain

| MP# | Completed Time of 72-hr storm (PST) | | 72-hr Total Basin Averaged Precipitation (mm) | |
| --- | --- | --- | --- | --- |
| | Small Domain | Big Domain | Small Domain | Big Domain |
| MP1 | January 3, 1997 0000 | January 2, 1997 1700 | 480.9 | 424.7 |
| MP2 | January 3, 1997 0000 | January 2, 1997 1800 | 380.0 | 427.5 |
| MP3 | January 3, 1997 0000 | January 2, 1997 0700 | 340.3 | 345.44 |
| MP4 | January 3, 1997 0000 | January 2, 1997 0800 | 313.5 | 330.5 |
| MP5 | January 2, 1997 1600 | January 2, 1997 1000 | 362.5 | 403.6 |
| MP6 | January 3, 1997 0000 | January 2, 1997 1000 | 336.3 | 368.6 |
| MP7 | January 3, 1997 0000 | January 3, 1997 0000 | 276.5 | 318.5 |
| MP8 | January 3, 1997 0000 | January 2, 1997 0800 | 293.9 | 331.6 |
| MP9 | January 2, 1997 1500 | January 2, 1997 2000 | 346.4 | 349.4 |
| MP28 | January 2, 1997 1600 | January 2, 1997 2000 | 373.4 | 363.1 |

*"The analysis in this study is inadequate to advance our understanding in model's physics. The manuscript lacks a conclusive assessment of the different MPs, as they behave quite diverse at different sites. The simple model-observation comparisons without any process-level diagnostics in this study fail to shed any light on the essential differences between MPs".*

I completely agree with your comment. Topical editor, Referee #2, and Referee #3 have also similar comments. Actually I left this part for another manuscript but I can see after your comments that this part is essential. If you would agree on this, I would like to add following part, which discuss model's physics more detail, to the final version of the manuscript.

**Precipitable Water**
The reason of New Year's flood is pineapple express, which is associated with a penetration of extensive amount of moisture merged in Hawaii to the West Coast of California. Precipitable water

is one of the good indicators of pineapple express which can be seen as a plume in Figure S2. Precipitable water variation in the domains of this study can be seen in Figure S3, in detail, for both NNRP Data (Left Panel) and ERA40 Data (Right Panel).

[Figure]

Figure S2. Pineapple express event in New Year of 1997 (Left Panel-NNRP Data; Right Panel ERA40 Data).

[Figure]

Figure S3. Precipitable Water variation in coarse WRF Domain for New Year of 1997 6am UTC (Left Panel-NNRP Data; Right Panel ERA40 Data).

[Figure]

Figure S3. Precipitable Water variation in fine WRF Domain for New Year of 1997 6am UTC (Left Panel-NNRP Data; Right Panel ERA40 Data).

Figure S2 and S3 are plotted by using NCEP/NCAR Reanalysis data (Left panels), which compose the initial and boundary conditions of the WRF model simulations of this study, and ECMWF ERA-40 Reanalysis data (right panels). These figures indicate that ERA40 Data has produced about 5-10kg/kg more columnar specific humidity than that of NCEP/NCAR Reanalysis data. This difference show us the importance of utilizing initial and boundary conditions as discussed in the conclusion section.

Spatial distributions of precipitable water for each evaluated MP schemes are presented in Figure S4 on January 1st, 1997 at 6 am UTC. The reason of choosing this date is that the most intense pineapple express penetration to the land occurred on this time which is followed by the extreme precipitation events occurred following hours depending on the local forcings. Although all 19 MP scheme's distributions look alike at first glance, the detailed visual analysis may indicate the differences and also similarities in MP's which are also grouped in section 3.2 MP schemes, accordingly. It can be concluded that each MP schemes are handled moisture distribution differently in time, depending on their microphysical processes, drop size distribution, terminal velocity formulations. This may result different moisture flux convergence rates. Thus, each MP scheme has resulted various precipitation amount and their precipitation timings also vary depending on the location.

[Figure]

Figure S4. Precipitable water variations in 19 MP schemes on January 1st, 1997 at 6 am UTC.

---

## Author Comment (AC2) · 1 Nov 2016

I would like to thank Anonymous Referee #3 for their time to evaluate my manuscript. I also appreciate their constructive evaluation style. Here are my replies, following Referee #3's suggestions in bold-italics.

***The study shows which microphysics (MP) schemes perform the best for this particular event. A key point (and one which would make this paper original, and significant) would be to find the common factors among the "successful" MP schemes, in other words, Why do these particular schemes perform the best? This "Why" is not addressed in the paper. Without this cause-and-effect relationship being established, it is difficult to know whether the current findings are applicable to other extreme events.***

Please see my reply to Referee #1 where I partly answer your comment. I tried to explain the reasons might affect these various results and why these particular schemes perform best leaving for future studies to be analyzed in detail. Since these differences are coming from the treatment of microphysical processes, these processes, such as number of variables, drop size distribution, terminal velocity formulations, should be discussed more detail. Moreover, according to my analyses, not shown here, precipitation amount values could be more reliable if moisture flux convergence approximation should have been used in these schemes for final precipitation, especially for extreme precipitation forecasting, because the estimation of both moisture amount and wind fields of the WRF model is much more reliable than that of microphysical processes. If you think that this approach must be added to the paper, I am willingly to do that. Thus, if I add this part to the paper, there would be no question that other extreme events can also be predicted with some accuracy level.

***At the beginning of the manuscript, the model simulations are de- scribed as having 28 levels. Later in the manuscript this appears to be 41 levels. Which is true? In either case, is this number of vertical levels sufficient to model events of this type? I think a few sentences here would be useful.***

The current study has 41 vertical levels and the study (Tan, 2010) that I compared to for basin-averaged values had 28 vertical levels. As a result, I showed that 41 levels would add more precipitation to the system because all MP schemes over-precipitated than that of Tan's study (2010) and the only difference with these two model structure is the vertical level number. Yes, generally 28 to 41 vertical levels would be compatible with the horizontal resolution of 3 km.

*P22, L25ff. "Model uncertainties..." I think the author is rather opening a "can of worms" here. It is not evident whether differences in model results associated with differing MP schemes are comparable with, or greater than, the differences in, say, horizontal resolution. This would represent a significant undertaking, however, with many degrees of freedom. I wonder if using WRF in some idealized framework may be one way to deal with this.*

I, especially, thank you for your comment on this issue. We may call this as my courage of ignorance but, actually, yes, I am trying to open *a "can of worms" here for future studies, so that we may start to think how we can initiate new approaches for MP schemes, rather than trying the combinations of current closure techniques. Precipitable water analyses that I have included to my replies to Referee #1, clearly states that effort which I would like to state in this paper that it may not work for point-wise analyses of extreme precipitation events. It is very true that model uncertainty identification studies should start with idealized cases before the real events.*

---

## Author Comment (AC3) · 1 Nov 2016

I would like to thank Anonymous Referee #2 for their time to evaluate the manuscript. I appreciate if you see my replies to Referee #1 and #3 which also cover your concerns. 8 rain-gauge observation stations are analysed in this study, but only 3 of them are presented. I have added some results of the other 5 stations to the replies of Referee #1. I have also included Precipitable Water analyses as a process level evaluation (Please see replies to Referee #1).